# Biparental contributions of the *H2A.B* histone variant control embryonic development in mice

Antoine Molaro[1]*[¤a], Anna J. Wood[1,2¤b], Derek Janssens[1], Selina M. Kindelay[1,2¤c], Michael T. Eickbush[1,2¤d], Steven Wu[1], Priti Singh[3], Charles H. Muller[4], Steven Henikoff[1,2], Harmit S. Malik[1,2]*

**1** Division of Basic Sciences, Fred Hutchinson Cancer Research Center, Seattle, Washington, United States of America, **2** Howard Hughes Medical Institute, Fred Hutchinson Cancer Research Center, Seattle, Washington, United States of America, **3** Comparative Medicine, Fred Hutchinson Cancer Research Center, Seattle, Washington, United States of America, **4** Male Fertility Laboratory, University of Washington School of Medicine, Seattle, Washington, United States of America

¤a Current address: Genetics, Reproduction and Development (GReD) Institute, Université Clermont Auvergne, Clermont-Ferrand, France
¤b Current address: Biochemistry, Molecular, and Cell Biology program, Cornell University, Ithaca, New York, United States of America
¤c Current address: Maggert Lab, Department of Cellular and Molecular Medicine, University of Arizona, Tucson, Arizona, United States of America
¤d Current address: Zanders Lab, Stowers Institute, Kansas City, Missouri, United States of America
* antoine.molaro@uca.fr (AM); hsmalik@fredhutch.org (HSM)

**Data Availability Statement:** All of the raw CUT&RUN data has been deposited at GEO, accession: GSE151639.

## Abstract

Histone variants expand chromatin functions in eukaryote genomes. *H2A.B* genes are testis-expressed short histone H2A variants that arose in placental mammals. Their biological functions remain largely unknown. To investigate their function, we generated a knockout (KO) model that disrupts all 3 *H2A.B* genes in mice. We show that *H2A.B* KO males have globally altered chromatin structure in postmeiotic germ cells. Yet, they do not show impaired spermatogenesis or testis function. Instead, we find that *H2A.B* plays a crucial role postfertilization. Crosses between *H2A.B* KO males and females yield embryos with lower viability and reduced size. Using a series of genetic crosses that separate parental and zygotic contributions, we show that the *H2A.B* status of both the father and mother, but not of the zygote, affects embryonic viability and growth during gestation. We conclude that *H2A.B* is a novel parental-effect gene, establishing a role for short H2A histone variants in mammalian development. We posit that parental antagonism over embryonic growth drove the origin and ongoing diversification of short histone H2A variants in placental mammals.

## Introduction

Histone proteins package eukaryotic genomes in nucleosomes, which are the basic unit of chromatin. This packaging function is essential for genome regulation and for the transmission of epigenetic information [1–3]. Nucleosomes are not only typically composed of the core

**Funding:** This work was funded by a postdoctoral fellowship DRG:2192-14 including full salary from the Damon Runyon Cancer Research Foundation (DRCRF) to AM, by R01 grant HG010492 from the National Human Genome Research Institute (NHGRI) at the National Institutes of Health (NIH) to SH, by R01 grant GM074108 from the National Institute of General Medical Sciences (NIGMS) at the National Institutes of Health (NIH) to HSM, and from the Howard Hughes Medical Institute (HHMI) Investigator grants, including full salaries to SH and HSM. The funders had no role in study design, data collection and analysis, decision to publish, or preparation of the manuscript.

**Competing interests:** I have read the journal's policy and the authors of this manuscript have the following competing interests: HSM is a member of the PLOS Biology editorial board.

**Abbreviations:** ADB, antibody dilution buffer; AutoCUT&RUN, automated Cleavage Under Target & Release Under Nuclease; chrX, X chromosome; CMA3, chromomycin A3; DSP, daily sperm production; FHCRC, Fred Hutchinson Cancer Research Center; gRNAs, guide RNAs; FRiPs, fraction of reads in peaks; HET, heterozygous; H&E, haemotoxylin and eosin; H-PAS, Hematoxylin-Periodic Acid Schiff; KO, knockout; MLH1, MutL homolog 1; MNase, micrococcal nuclease; ORF, open reading frame; PRDM9, PR domain containing protein 9; RNA-seq, RNA sequencing; SCP3, synaptonemal complex protein 3; SEACR, sparse enrichment analysis for CUT&RUN; TALENs, transcription activator-like effector nucleases; tracrRNA, trans-activating crispr RNA; TUNEL, terminal deoxynucleotidyl transferase dUTP nick end labeling; UCSC, University of California, Santa Cruz; WT, wild-type.

histone proteins H2A, H2B, H3, and H4, but also can include noncanonical histone variants, which have specialized functions during cell division, gene expression, and development [3–5]. Like the genes encoding core histones, most histone variant genes originated in the last common ancestor to all eukaryotes and are highly conserved [6–8].

In contrast to other histone genes, short H2A histone variants are both evolutionarily young and rapidly evolving. They originated on the X chromosome (chrX) of the last common ancestor of placental mammals and diversified into 4 phylogenetically distinct clades: *H2A.B*, *H2A.L*, *H2A.P*, and *H2A.Q* [9]. These clades of histone variants are subject to high gene turnover and positive selection [9]. In addition to their unusual evolution, short H2A variants also have atypical biochemical properties. In vitro and ectopic expression studies showed that nucleosomes containing H2A.B are more labile and associate with regions of high nucleosome turnover [10–18]. Despite their unusual evolution and functions, the biological roles of short H2A variants have been mysterious until recently.

Short H2As are predominantly expressed during male spermatogenesis [9,15,19,20]. Recent in vivo studies of H2A.Bs and H2A.Ls have begun to reveal insights into their function. In mice, H2A.Ls accumulate at the exit of meiosis, and some are retained in the sperm, but eventually disappear from the paternal pronucleus following fertilization [15,20–22]. The knockout (KO) of a single mouse-specific *H2A.L* gene on chromosome 2 results in male infertility with failure to initiate histone-to-protamine replacement [23]. H2A.B accumulates on the chromatin of meiotic and postmeiotic cells in mice [15,24,25]. A recent study deleted all 3 copies of *H2A.B* on mouse chrX using transcription activator-like effector nucleases (TALENs) [26]. Resulting KO males displayed defects in the chromatin organization and splicing of active genes in postmeiotic cells, but only had subtle defects in fertility. Thus, the biological functions of *H2A.B* that would explain their long-term retention in most placental mammals remain largely uncharacterized.

To gain further insight into *H2A.B* function, we focused on their unusual evolutionary dynamics. We considered the evolutionary hypothesis that *H2A.B*'s diversification might be the consequence of an ongoing genetic conflict shaping the reproduction of placental mammals [9]. We considered 2 potential sources of genetic conflicts. Based on the X-linkage of *H2A.B* in all placental mammals and its expression during male meiosis, we first hypothesized that *H2A.B* might be involved in meiotic drive. Under this scenario, *H2A.B* genes might act as selfish genetic elements that promote the transmission of the chrX carrying them. X-versus-Y meiotic drive has been shown to influence sperm quality (e.g., motility) and lead to imbalanced sex ratios in litters [27–32].

Alternatively, we hypothesized that *H2A.B* might function postfertilization in parental conflicts akin to genomic imprinting [33,34]. Such conflicts arise from asymmetric maternal and paternal resource allocation to the progeny and lead to parental antagonism over genes that regulate these processes [33,34]. Such conflicts are particularly important in placental mammals where gestation occurs entirely in utero [35]. Consequently, imprinting defects result in reduced embryo fitness and poor implantation outcomes but not necessarily impaired fertility [33–37].

To test these alternate evolutionary hypotheses, we investigated the reproductive outcomes of *H2A.B* triple KO (*ΔH2A.B*) mice generated using a CRISPR/Cas9-mediated genome-editing strategy. Using chromatin profiling in spermatids, we showed that the loss of H2A.B results in altered chromatin structure consistent with genome-wide nucleosome unwrapping. Yet, these changes did not alter spermatogenesis or affect sex bias of progeny. Instead, we found that the frequency of inviable conceptuses is increased in crosses between *ΔH2A.B* males and *ΔH2A.B* females. Moreover, surviving embryos from crosses between *ΔH2A.B* parents have reduced fetal and placental weights compared with wild-type (WT) crosses. Using genetic crosses that

separate parental contributions from zygotic genotypes, we show that both paternal and maternal contributions of *H2A.B* affect this phenotype. We further show that female embryonic germ cells also express *H2A.B* at the onset of meiosis I. We conclude that *H2A.B* acts as a parental-effect gene that controls postimplantation embryo development. Thus, *H2A.B* is the first reported biparental-effect histone variant gene in mammals. Our work supports the hypothesis that parental antagonism drives the rapid diversification of this histone variant in placental mammals [38–40].

## Results

### Generation of *ΔH2A.B* mice via CRISPR/Cas9 editing

The mouse chrX encodes 3 intact *H2A.B* genes (S1 Fig) and 1 pseudogene, which shares approximately 86% to 89% identity with other *H2A.B* genes but contains multiple frameshift mutations that disrupt its reading frame [9]. Two of the intact *H2A.B* genes (*H2afb1* and *H2afb2*) are located in a "head-to-tail" arrangement in a large intergenic region, whereas the third (*H2afb3*) is located in the reverse orientation, approximately 4-Mb downstream, within an approximately 73-kb intron in the 5′ UTR of *Pcdh11x* (UCSC genome browser [41]).

Given that all 3 *H2A.B* genes encode highly similar proteins (>80% identical) and therefore might function redundantly [9], we aimed to disrupt all 3 genes to generate a triple KO mouse strain. For this, we performed CRISPR/Cas9 editing by pronuclear injection into *B6;SJLF1/J* zygotes (S1 Fig). We targeted guide RNAs (gRNAs) to 2 sites found to be 82 bp apart in highly homologous regions of all 3 *H2A.B* genes (S1 and S2 Figs). These experiments yielded 6 founders with germline transmission, with genomic alterations in 1, 2, or all 3 *H2A.B* genes. We identified the molecular lesions by Sanger sequencing and focused on 1 strain that had mutations in all 3 *H2A.B* genes. Two genes had the targeted 82-bp deletion that led to loss of nearly two-third of the protein-coding open reading frame (ORF), whereas the third gene had 2 small (15 bp and 5 bp) deletions that created a 5 aa deletion and a frameshift mutation leading to loss of nearly half the ORF (S1 and S2 Figs). We refer to this triple KO strain as the *ΔH2A.B* or KO line.

In order to ensure relatively isogenic genetic backgrounds, we backcrossed heterozygous (HET) *ΔH2A.B/+* females to "pure" *C57BL/6J* males for between 5 and 8 generations. SNP genotyping showed that this backcrossing strategy yielded *ΔH2A.B* animals that were over 92% pure *BL/6J* at backcross 7 (S3 Fig). In the course of these backcrosses, we generated hemizygous *ΔH2A.B/Y* (KO) and WT (*+/Y*) male siblings as well as homozygous (*ΔH2A.B/ΔH2A.B*, KO) and HET (*ΔH2A.B/+*) female siblings. The close genomic proximity of 3 *H2A.B* genes lowers the likelihood of a meiotic recombination event between these genes. However, to rule out the possibility that a rare recombination event in HET females could separate the KO alleles, we constantly monitored the inheritance of all 3 KO alleles through all subsequent experiments via genotyping (S1 Fig).

### *H2A.B* affects chromatin structure in postmeiotic male germ cells

*H2A.B* has been reported to be primarily expressed during male meiosis [9,15,24,42]. We, therefore, investigated whether the loss of *H2A.B* resulted in changes in chromatin structure in male germ cells. We isolated live haploid spermatids from KO and WT adult testes using Hoechst staining and propidium iodide cell sorting [43]. We then used automated Cleavage Under Target & Release Under Nuclease (AutoCUT&RUN), an antibody-targeted nuclease cleavage method, for high-resolution chromatin profiling (Fig 1, S4 Fig) [44]. To assess the overall status of chromatin genome-wide, we profiled 3 histone modifications in distinct chromatin compartments: H3K4me3 (typically associated with active promoter regions), H3K9me

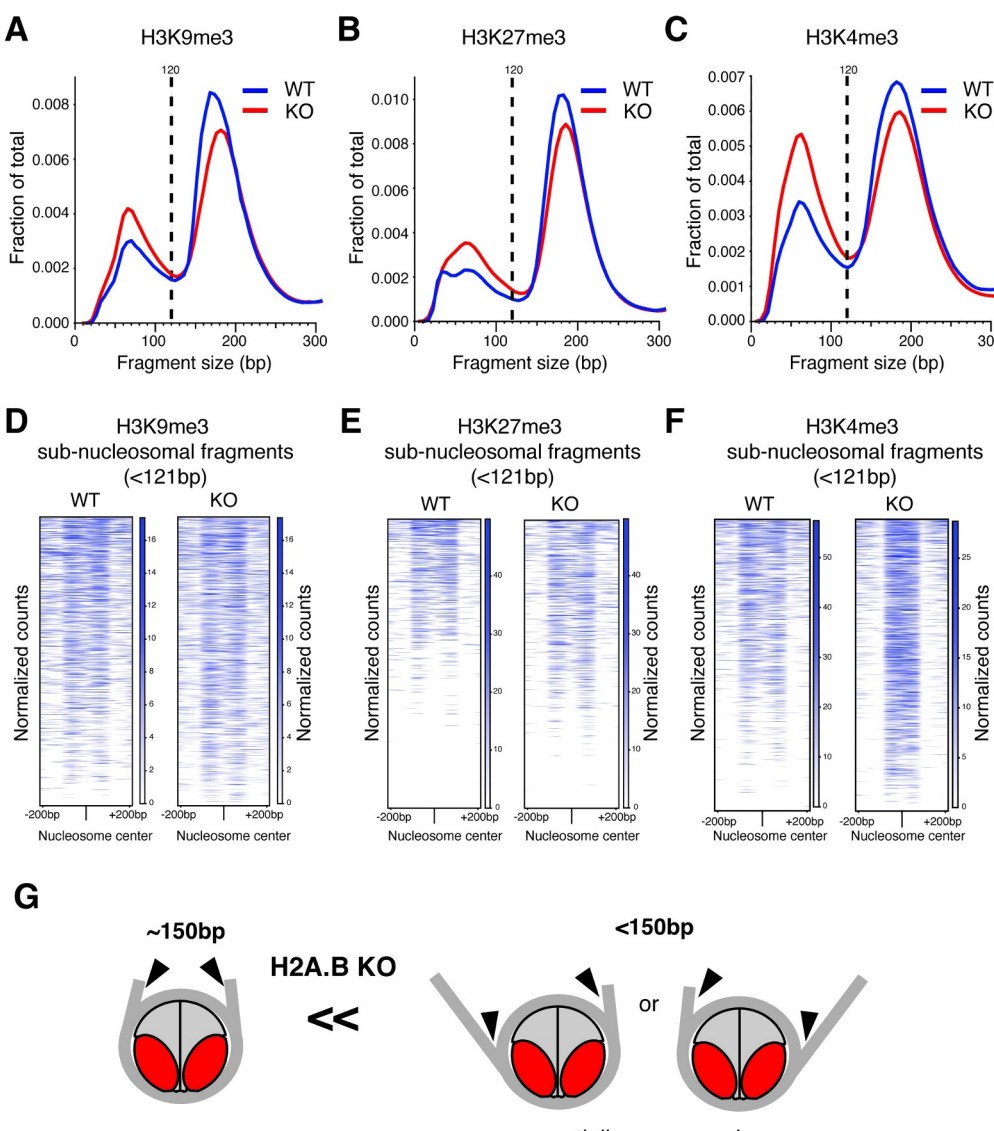

**Fig 1. Chromatin landscapes of haploid male germ cells.** Size distribution of fragment released by AutoCUT&RUN in spermatid cells from WT (blue) and *H2A.B* triple KO (red) mice using antibodies for H3K9me3 (A), H3K27me3 (B), and H3K4me3 (C) histone modifications. To allow comparison between conditions, we down-sampled mapped fragments to be identical between WT and KO samples (see S1 Table and S1 Data). Fragment sizes corresponding to sub-nucleosomal fragments are to the left of the 120-bp mark (dashed line). (D, E, F) Heatmap of normalized counts (blue scale) of short fragments (<121 bp) over the top 2,000 low fuzziness nucleosomes in each sample. Regions are centered on nucleosomes profiled with H3K9me3 (D), H3K27me3 (E), and H3K4me3 (F). (G) Schematic of partial nucleosome unwrapping leading to the formation of long or short fragments by AutoCUT&RUN (arrowheads indicate MNase cleavage). Based on these findings, we conclude that *H2A.B* KO samples have a larger fraction of partially unwrapped nucleosomes. KO, knockout; WT, wild-type.

(enriched at sites of constitutive heterochromatin), and H3K27me3 (marks Polycomb-repressed domains). For each of these histone marks, fragment coverage between WT and KO samples was almost identical throughout the genome or in all called peaks (see Materials and methods) with no significant differences in fragment distribution between autosomes and X or Y chromosomes (S4 Fig). These results suggest that *H2A.B* loss does not alter the global distribution of these chromatin domains in postmeiotic germ cells.

CUT&RUN relies on processive cleavage by tethered micrococcal nuclease (MNase), which releases DNA fragments that can be sequenced to provide detailed footprints of the antibody-targeted proteins [45–47]. For histone modifications like H3K4me3, H3K9me3, or H3K27me3, CUT&RUN recovers both nucleosome-sized DNA fragments ($\geq$150 bp) as well as sub-nucleosomal DNA fragments ($\leq$120 bp) (S5 Fig). These sub-nucleosomal fragments correspond to either partially unwrapped nucleosomes or neighboring DNA bound by proteins such as transcription factors [46,48,49]. Surprisingly, we found that CUT&RUN analyses on *H2A.B* KO spermatids released relatively fewer mono-nucleosome–sized fragments and more sub-nucleosomal fragments than WT across all 3 chromatin domains (Fig 1A–1C). In all 3 cases, the relative proportions of sub-nucleosomal to mono-nucleosome fragments were statistically higher in *H2A.B* KO spermatids (S1 Table). This indicates that H2A.B either affects nucleosome wrapping or accessibility in most chromatin compartments.

To distinguish between these possibilities, we analyzed the distribution of small fragments over well-positioned nucleosomes (also referred to as "low fuzziness" nucleosomes; see Materials and methods). We found that sub-nucleosomal fragments mapped to either the 5′ or 3′ edges extending toward the center of nucleosomes in both WT and KO samples (Fig 1D–1F, S6 Fig). This distribution is consistent with sub-nucleosomal fragments arising via nucleosome unwrapping, which allows MNase digestion internal to the nucleosome-protected DNA [48] (Fig 1G). Sub-nucleosomal fragments released by H3K9me3 and H3K27me3 nucleosomes were similarly distributed between WT and KO. In contrast, sub-nucleosomal fragments released by H3K4me3 nucleosomes mapped further toward the center of the nucleosome in KO spermatids (Fig 1F). These sub-nucleosomal H3K4me3 fragments were strongly enriched over transcriptional start sites (S6D Fig). We attribute these differences to the unique positioning and high rate of turnover of H3K4me3 nucleosomes at promoters making them already highly prone to unwrapping during transcription [5]. Our findings support prior observations that regions of high nucleosome turnover appear to be especially sensitive to *H2A.B* loss [10,18,24–26].

Next, we examined whether the changes we observed in the ratio of sub-nucleosomal- to nucleosome-sized fragments occurred at specific sites in the genome or instead reflect global changes in the chromatin structure of *H2A.B* KO spermatids. First, we compared the fraction of reads in peaks between differently sized fragments (fraction of reads in peak [FRiPs]) (S2 and S3 Tables). We found that both sub-nucleosomal and nucleosome fragments had similar FRIPs (S2 Table), indicating that these occurred in similar genomic regions. Next, we computed the ratios of small (<121 bp) versus large (>149 bp) fragments in peaks. If the loss of *H2A.B* produced more sub-nucleosomal fragments in specific peaks, we would expect these changes to be consistently present across *H2A.B* KO replicates. Contrary to this expectation, while all *H2A.B* KO samples produced more sub-nucleosomal fragments than the WT samples, analysis of the peaks showed that KO:WT enrichments were not the same across replicates (S6E Fig). Consistent with previous reports of H2A.B being broadly distributed in the nucleus [24,25], we conclude that *H2A.B* loss leads to widespread genome-wide nucleosome unwrapping rather than at specific loci.

Finally, we noted that mono-nucleosome–sized fragments released by CUT&RUN were larger in the KO than in the WT (Fig 1A–1C). Thus, mono-nucleosomes with smaller footprints than "canonical" size are absent in *H2A.B*-KO animals. We hypothesize that these nucleosomes with smaller footprints correspond to H2A.B nucleosomes in WT animals, which are missing in KO animals (S5 Fig). Our findings are consistent with previous in vitro studies, which showed that H2A.B-containing mono-nucleosomes have smaller (by approximately 30 bp) MNase footprints than those with core H2A [11,12,17].

## *ΔH2A.B* does not impair spermatogenesis

If H2A.B's chromatin function played a major role during germ cell development, we would expect KO males to have altered spermatogenesis and reduced fertility. To investigate whether *H2A.B* loss led to impaired germ cell production, we performed histological analyses of testis cross sections. These analyses did not reveal any significant differences in cell types (e.g., all germ cell types could be readily identified) or seminiferous tubule organization between WT and KO males (S7A Fig). Moreover, the rates of production and motility of sperm from KO males were not significantly different from WT males (S7B and S7C Fig). We also investigated sperm morphology from WT and KO males. Over 85% of sperm cells displayed the expected normal morphology in both samples (S7D Fig). Although the KO samples displayed a slightly increased variance in the proportion of abnormal sperm heads, on average, WT and KO samples had overlapping distributions of frequencies of abnormal heads, necks, and tails (S7D Fig). Thus, we found no signs of impaired progression through spermatogenesis or major defects in sperm development in KO males.

We hypothesized that the altered chromatin structure of germ cells in KO males might affect germline genome integrity. We first investigated DNA damage during meiosis I, when H2A.B is first incorporated into chromatin [24–26]. We did not find any significant differences in the frequency of terminal deoxynucleotidyl transferase dUTP nick end labeling (TUNEL)–positive (which mark single-strand DNA breaks) seminiferous tubules between KO and WT males (S8A Fig). In both cases, TUNEL-positive cells were located toward the basal side of the tubules corresponding to cells entering meiosis I. Next, we measured crossover frequencies and gammaH2A.X incorporation (a marker of double-strand DNA breaks) in meiotic germ cell spreads. Again, we found no significant differences between WT and KO germ cells (S8B and S8C Fig). We also investigated genome integrity in epididymal sperm. We subjected epididymal sperm from 3 pairs of age-matched sibling WT and KO males to both alkaline diffusion and neutral comet assays, 2 independent methods to assess genome integrity (see Materials and methods). Although we found increased numbers of damaged sperm in some KO males relative to their age-matched WT siblings in the alkaline diffusion assays, these differences were not consistent when compared with the comet assay or when compared across all cases (S8D Fig). Overall, based on these findings, we conclude that *H2A.B* loss does not significantly impair genomic integrity either during meiosis or in mature gametes.

Our chromatin profiling analyses had suggested that the loss of H2A.B might affect global chromatin compaction in postmeiotic male germ cells. We investigated whether these translated to defects in sperm genome compaction. For this, we used chromomycin A3 (CMA3) staining, which intercalates with DNA when sperm nuclei have reduced protamine content. Thus, increased CMA3 staining is an indirect measure of overall genome deprotamation in sperm [50]. We counted 200 sperm cells per male across 2 independent pairs of age-matched WT and KO siblings. We found that KO samples had fewer CMA3 positive cells than WT (WT = 9% versus KO = 0.5% and WT = 3% versus KO = 1%, S8E Fig). Our findings rule out a gross deprotamation defect upon the loss of H2A.B. Instead, they suggest that sperm in H2A.B KO sperm have more compact chromatin than that in WT sperm, consistent with increased rather than decreased protamation.

## Loss of H2A.B impairs male reproductive fitness

To assess the biological role of *H2A.B*, we tested the effect of *H2A.B* loss in a fertility assay [51, 52]. For this, we mated sibling WT (*n* = 4) or KO (*n* = 6) to age-controlled *C57BL/6J* females, allowing just 1 mating per male–female pair (see Materials and methods). These crosses yielded a total of 28 and 37 litters sired from WT and KO males, respectively (S4 Table). We

found no deviations from Mendelian expectations in sex ratios between litters from WT and KO males in these litters (S4 Table and S9A Fig). We also found no evidence of skewed sex ratios in our long-term breeding experiments (S9B Fig). Based on these findings, and the fact that we found little perturbation during gametogenesis, we conclude that there is no indication that *H2A.B* participates in X-versus-Y meiotic drive [27,30,31,53].

When comparing litter sizes between KO and WT males, we observed that KO sires produced more small litters than their WT counterparts (Fig 2A), resulting in a non-normal distribution of litter sizes ($p = 0.0002$, Shapiro–Wilk test, S4 Table). However, this effect did not

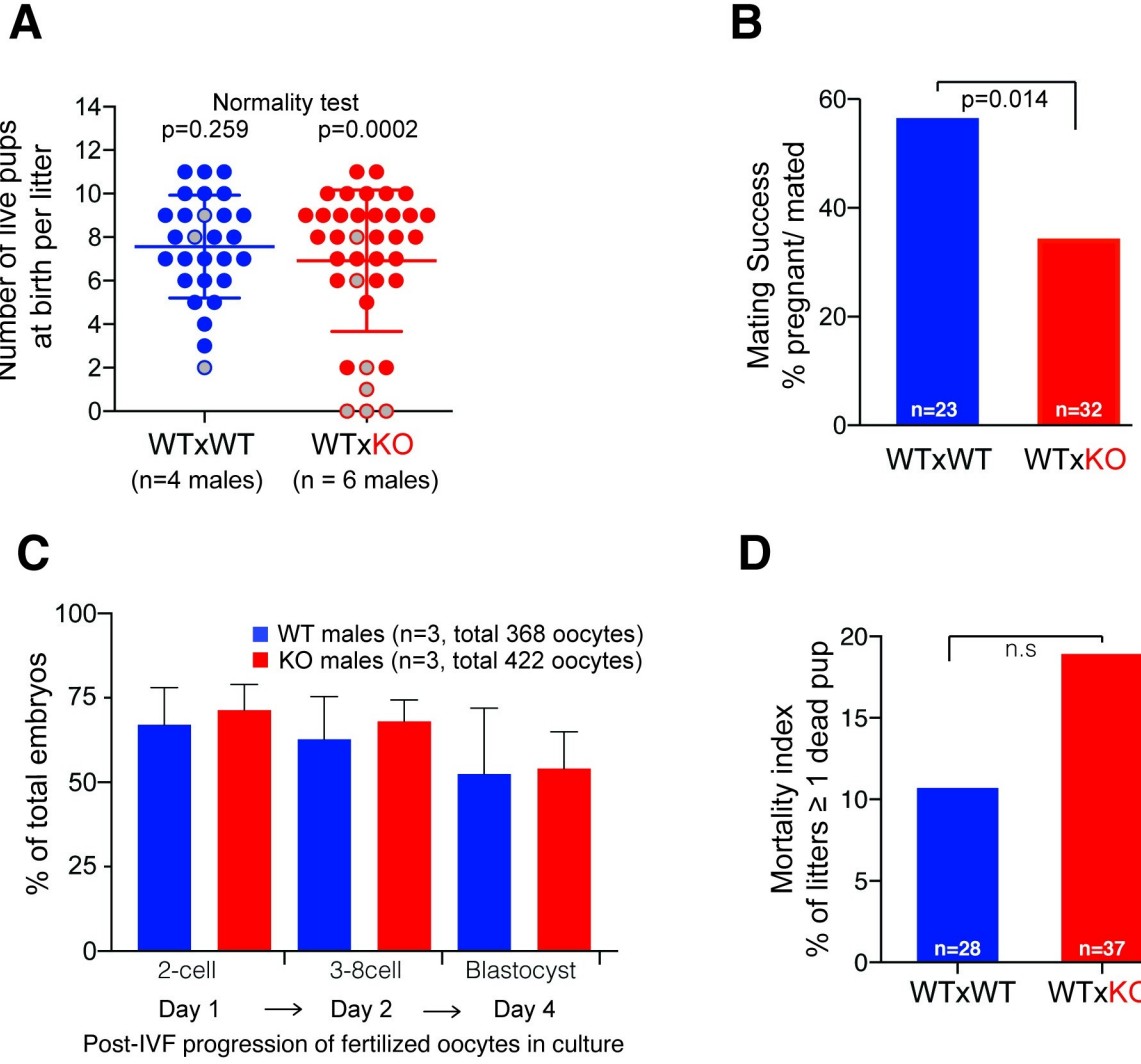

**Fig 2. Subfertility of H2A.B KO males.** (A) Distribution of litter sizes (live pups) of WT ($n = 4$, blue circles) and KO ($n = 6$, red circles) males over 9 months. Gray-filled circles correspond to litters with at least 1 dead pup post-birth. Results of Shapiro–Wilk tests of normality are shown at the top. (B) Proportion of successful matings of WT (blue) and KO (red) males, calculated as the percentage of females with a reproductive plug ("mated") that became pregnant. Total number of females is shown at the bottom of each bar. *p*-value is based on a chi-squared test (KO vs. expected based on WT data). (C) Percentage of embryos that progressed to the 2 cell, 3–8 cell, or blastocyst stages, respectively, at day1, 2, and 4 post–in vitro fertilization of WT oocytes with KO (blue bars) or WT (red bars) sperm. Mean and standard deviation are shown across 3 replicate experiments (see Materials and methods). (D) Fraction of litters with at least 1 dead pup (mortality) for WT (blue) and KO (red) males. Total number of litters is shown at the bottom of each bar. Mating scheme is indicated at the bottom: dam × sire. Raw data can be found in S4 Table and S2 Data. IVF, *in vitro* fertilization; KO, knockout; WT, wild-type.

lead to significant differences between the average litter size produced by WT or KO sires (2-tailed Mann–Whitney test, S4 Table, Fig 2A). Our results are qualitatively similar to a recently published study that performed a *H2A.B* KO using TALENs in an FVB/NJArc mouse genetic background; however, that study found a significant albeit subtle effect of *H2A.B* loss on mean litter sizes, whereas we did not [26].

We next examined whether other aspects of male reproductive fitness were impaired upon loss of *H2A.B*. Therefore, we measured the mating and fertilization success of KO versus WT male mice. Mating by male mice leads to the deposition of a reproductive "plug", which delays re-mating by other males in a natural setting. We found that KO males were able to deposit reproductive plugs in receptive females at similar rates as WT males, suggesting that mating frequency was not impaired by *H2A.B* loss (S4 Table). However, matings by KO males resulted in pregnancies in only 34% of cases, compared with 57% for WT ($p = 0.014$, chi-squared test, Fig 2B). This result indicates that loss of H2A.B significantly decrease males' mating success.

Decreased mating success could result from impaired fertilization or failure to undergo proper preimplantation or postimplantation development. To distinguish between these possibilities, we followed the progression of zygotes produced from WT oocytes that were in vitro fertilized with sperm from either WT or KO males ($n = 3$, >100 eggs/replicate/genotype; see Materials and methods). We found that a similar proportion of fertilized oocytes progressed to the 2-cell stage, whether the sperm was from WT or KO males (mean WT = 66% versus KO = 71%, Fig 2C). Thus, the loss of *H2A.B* does not impair the sperm's ability to successfully fertilize oocytes. Moreover, we found no significant difference in the progression of these embryos up to the blastocyst stage (Fig 2C). Thus, zygotes produced from KO males appear fully capable of developmental progression, at least *in vitro*. These findings suggest that the lower mating success of KO males results from defects arising postimplantation rather than preimplantation.

Consistent with this possibility, litters sired by H2A.B KO males exhibited an increase in the frequency of at least 1 dead pup at birth in fertility assays (WT = 10% versus KO = 18%, $p = 0.1$, chi-squared test, Fig 2D) or in long-term breeding experiments (WT = 22% versus KO = 26%, $p = 0.6$, 2-tailed chi-squared test with Yates correction, S9C Fig). This effect is consistent with our observation of non-normal distribution of litter sizes sired by KO males (Fig 2A).

In sum, we conclude that the loss of all 3 testis-expressed *H2A.B* genes does not appear to have a significant effect on sex ratios, spermatogenesis, or testes function in mice. Yet, *H2A.B*–KO males display fertility defects that seem to manifest postimplantation and affect perinatal viability. These findings rule out the meiotic drive hypothesis for *H2A.B* but are consistent with the alternative possibility that a parental genetic conflict may shape *H2A.B* function and evolution in placental mammals.

## *H2A.B* impacts embryo viability and fitness

To investigate *H2A.B*'s potential function in postimplantation development, we evaluated the rates of successful implantations at embryonic day 14.5 (E14.5) in series of crosses, starting with crosses between WT females and KO males. We found that there was a mild but nonsignificant increase in the fraction of resorbed or developmentally delayed conceptuses when compared with WT × WT crosses (3.7% versus 2.3%, Fig 3A). In the reciprocal cross between KO females and WT males, we also found no measurable effect on embryo viability (1.4% versus 2.3%, Fig 3A). Unexpectedly, however, in crosses between KO females and KO males, the fraction of resorbed or developmentally delayed conceptuses increased about 4-fold (9.6% versus 2.3%, $p = 0.0001$, chi-squared test, Fig 3A). Thus, 10% of *ΔH2A.B* zygotes from KO × KO crosses do not survive past implantation.

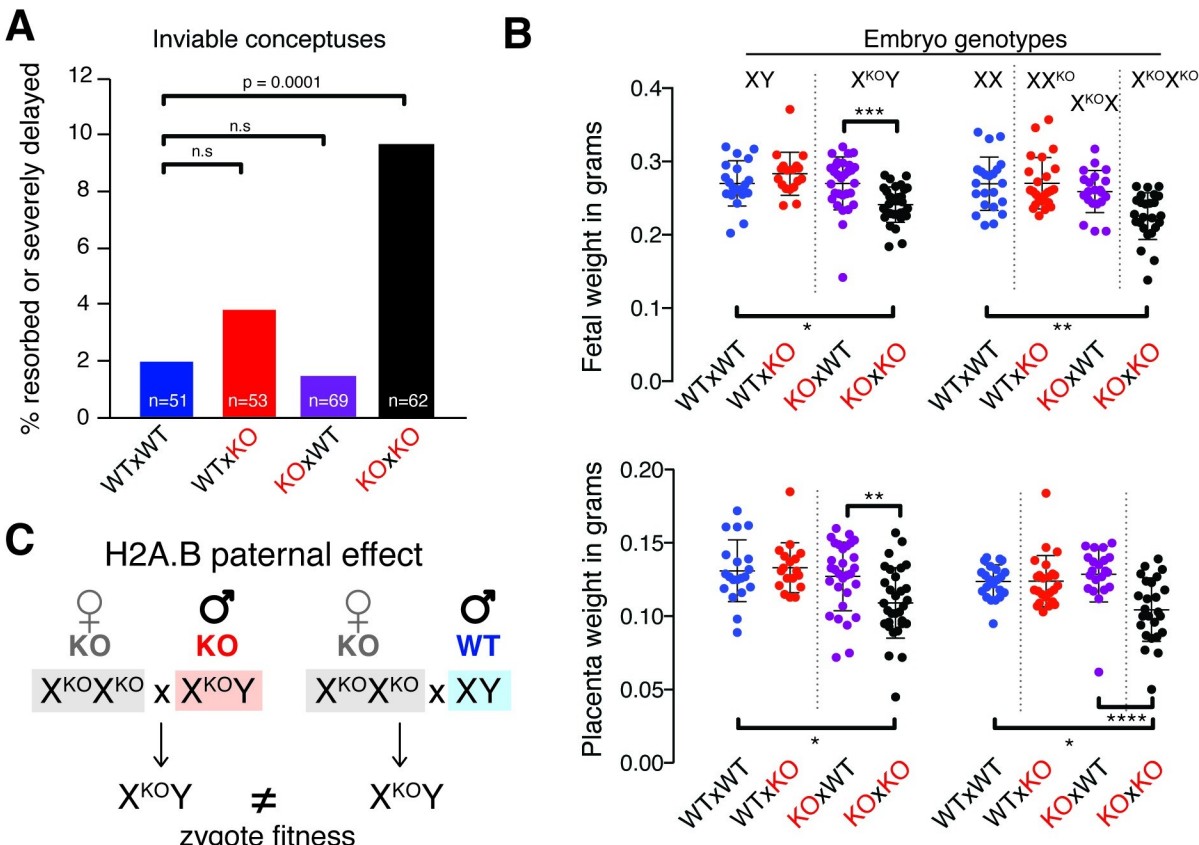

**Fig 3. Paternal effect of H2A.B KO on embryo fitness.** (A) Fraction (%) of severely delayed or fully resorbed conceptuses dissected at embryonic day 14.5 (E14.5) (also see S3 Data). Total number of pups for each cross type is indicated at the bottom of each bar. Bars are colored according to each parental cross. *p*-value is based on a chi-squared test (vs. expected WT). Mating schemes are indicated as dam × sire. (B) Embryo (top) and placenta (bottom) weight distributions for male (left) and female (right) conceptuses (also see S4 Data). Parental cross is shown at the bottom and embryo genotype is shown at the top of each graph. $X^{KO}$ indicates H2A.B KO chrX. Dots are colored according to parental crosses. Significance of Kruskal–Wallis ANOVA test is shown * $p < 0.01$, ** $p < 0.001$, *** $p < 0.0001$, and **** $p < 0.00001$. (C) Schematic of H2A.B paternal effect on embryo weights. Mating schemes are indicated as dam x sire. chrX, X chromosome; KO, knockout; WT, wild-type.

The second striking effect of H2A.B on embryonic fitness became evident upon analyzing surviving embryos. We found that fetal weights of both male and female conceptuses from KO × KO crosses were approximately 30% smaller than those from WT × WT controls or those from KO × WT or WT × KO reciprocal crosses (Fig 3B, Kruskal–Wallis ANOVA test for males or females, $p < 0.0001$). This reduction was also observed for placental weights, although with larger standard deviations (Fig 3B, Kruskal–Wallis ANOVA test, males $p = 0.0012$ and females $p < 0.0001$). Therefore, we infer that the reduced weights of KO × KO embryos might result from growth restriction. Overall, our results imply that *H2A.B*'s primary biological role appears to be in postfertilization development; embryos have reduced viability and impaired fitness postimplantation in the complete absence of *H2A.B*.

## Paternal effect of H2A.B on development

We considered 2 possible causes of the defects in embryos of KO × KO crosses. In the first model, the defects we observed could be entirely the result of zygotic deficiency. If this were the case, it would imply that zygotic *H2A.B* is required for early embryonic development.

Indeed, an earlier study found that *H2A.B* is expressed during the early cleavage stages of mouse embryogenesis [54]. Alternatively, these defects could be the result of missing parental contributions that are required for optimal embryo development in mice.

We decided to distinguish between these 2 possibilities by comparing zygotes with identical genotypes, but from genetically distinct parents. We first tested for paternal effects by generating genetically identical KO male embryos ($X^{KO}Y$, where $X^{KO}$ denotes a *ΔH2A.B* chrX) via crosses between KO females to either WT or KO males (Fig 3C). If the defects in embryo weights were the result of zygotic deficiency, we would expect $X^{KO}Y$ embryos to have the same lower weight in both crosses. Instead, we found that the fetal and placental weights of $X^{KO}Y$ males produced from WT fathers were significantly higher than those produced from KO fathers (Fig 3B, Kurskal–Wallis ANOVA, multiple comparison $p = 0.0004$). Since the maternal genotypes are identical (KO) in these crosses, we conclude that this difference must solely be due to the different paternal *H2A.B* genotypes. We thus conclude that *H2A.B* is a paternal-effect gene [35,55,56]. While some histone variants exhibit maternal effects [57], *H2A.B* is, to our knowledge, the first identified paternal-effect histone variant in mammals.

## Maternal contribution of H2A.B to development

Next, we investigated whether maternal *H2A.B* genotypes also influences embryonic development. For this, we crossed 3 possible female genotypes (WT, HET, or KO) to KO males. To ensure comparisons were influenced by maternal but not zygotic H2A.B genotypes, we focused on identical zygotic genotypes produced from crosses between genetically distinct mothers (e.g., HET versus KO) and fathers with identical KO genotype. For example, both the HET × KO and KO × KO crosses (dam × sire) produce $X^{KO}X^{KO}$ females and $X^{KO}Y$ males. We found no significant difference when comparing fetal or placental weights of these zygotes from the 2 crosses (Fig 4A).

However, when we compared XY males produced from WT × KO and HET × KO crosses, we observed a dramatic difference (Fig 4A). XY fetuses from HET × KO crosses were significantly smaller than XY fetuses from a WTxKO cross (Fig 4A). We observed a similar pattern for $X^{KO}X$ female embryos, which were larger when produced from WT females. Unlike in the paternal-effect crosses, we did not find significant differences in placental weights in multiple comparisons between these crosses (Fig 4A). Based on these results, we conclude that there is a significant maternal contribution of *H2A.B* to embryonic development, with WT mothers producing significantly larger fetuses than HET or KO mothers. In addition, based on the similarities between HET and KO maternal effects, we further infer that maternal *H2A.B* contribution to embryos size might be haplo-insufficient.

Having established that both paternal and maternal contributions of *H2A.B* shape embryonic development, we finally assayed whether zygotic *H2A.B* genotypes also influenced embryo development. To test this possibility, we reassessed fetal and placental weights of genetically distinct male (XY versus $X^{KO}Y$) and female ($XX^{KO}$ versus $X^{KO}X^{KO}$) embryos produced from genetically identical parents in a HET × KO crosses (Fig 4B). Since the maternal and paternal genotypes are identical in these comparisons, we reasoned that we would expect to see difference between XY and $X^{KO}Y$ embryos if embryo fitness was affected by zygote genotype (Fig 4B). However, we found no statistical difference in the weight of XY compared with $X^{KO}Y$ or $XX^{KO}$ compared with $X^{KO}X^{KO}$ embryos (Fig 4A, Kruskal–Wallis ANOVA test). Based on these crosses, we conclude that paternal and maternal contributions of *H2A.B* affect embryo development, whereas zygotic *H2A.B* genotype does not.

Our finding that *H2A.B* is also a maternal-effect gene is highly unexpected because most studies to date have shown *H2A.B* to be predominantly expressed in testes across a wide variety

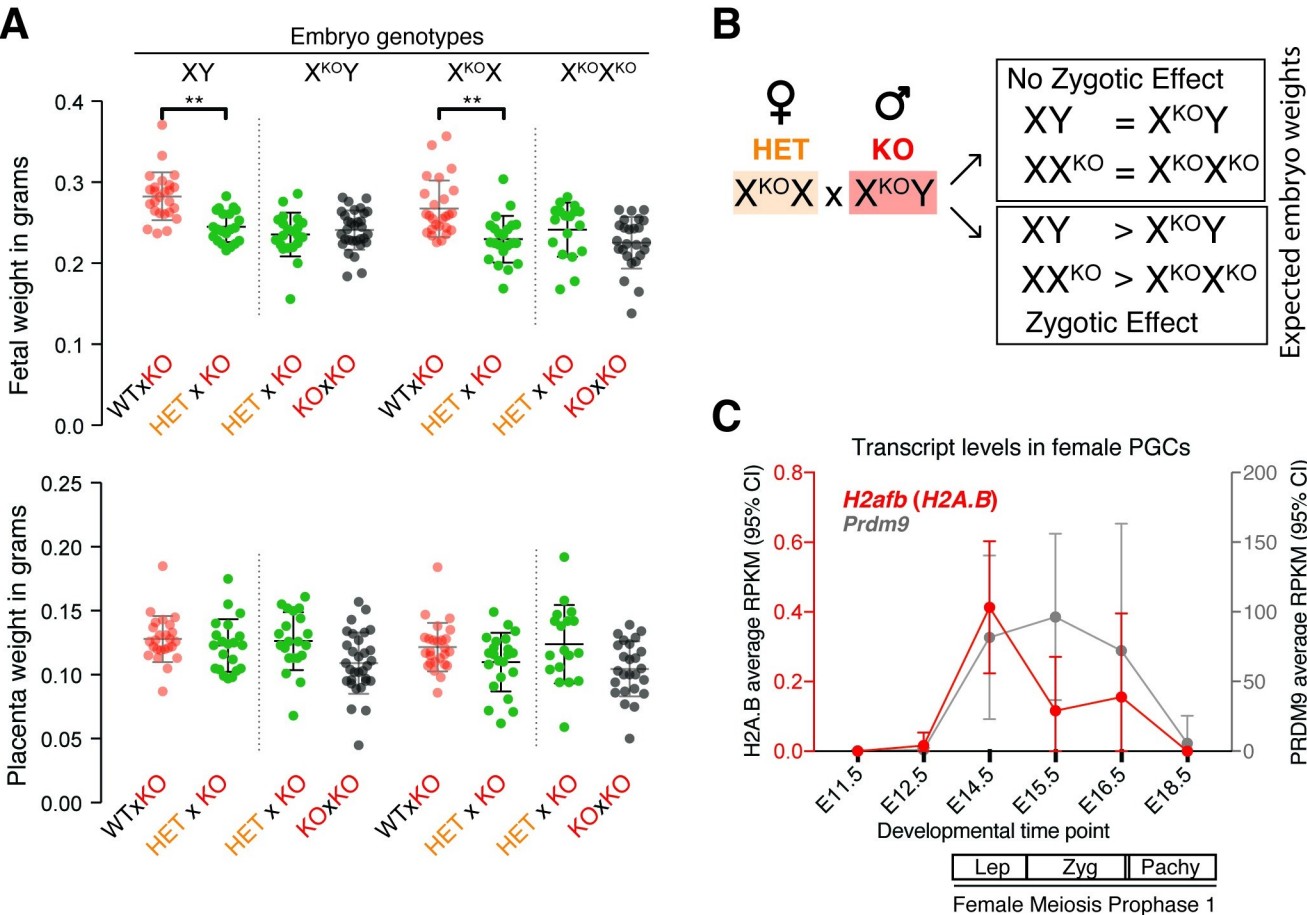

**Fig 4. Maternal effect of H2A.B KO on embryo fitness.** (A) Embryo (top) and placenta (bottom) weight distributions for male (left) and female (right) conceptuses from HET × KO (dam × sire) crosses (also see S4 Data). WT × KO and KO × KO data are from Fig 3B. Embryo genotype is shown at the top of each graph. $X^{KO}$ indicates H2A.B KO chrX. Dots are colored according to parental crosses. Statistically significant results ($p < 0.01$) of multiple comparisons Kruskal–Wallis ANOVA test are also shown, ** $p < 0.001$. (B) Schematic showing the expected outcomes of the presence (top box) or absence (bottom box) of a zygotic effect of H2A.B on embryo weights produced by a HET × KO cross. (C) RPKM values for *H2afb* (combining reads mapping to any of the 3 paralogs) (red, left y-axis) and *Prdm9* (gray, right y-axis) in RNA-seq from isolated female primordial germ cells from E11.5 to E18.5 [58] (also see S4 Data). Average RPKM and 95% CIs are shown for each time point. Annotation of the first meiotic prophase is shown at the bottom. CI, confidence interval; HET, heterozygous; KO, knockout; Prdm9, PR domain containing protein 9; RNA-seq, RNA sequencing; RPKM, reads per kb per million; WT, wild-type.

of mammals and some minor brain expression in mouse [9,15,24,42]. Yet our findings predict that *H2A.B* must also be expressed in females. The absence of a highly specific antibody to H2A.B prevented us from testing for the presence of H2A.B in oocyte nucleosomes. We, therefore, turned to RNA sequencing (RNA-seq) analyses to investigate *H2A.B* expression during oogenesis. We reasoned that *H2A.B* may follow a similar expression pattern in females as it does in males. If this were the case, expression should be detectable by the leptotene stage of meiosis I, which begins around E14.5 in the female genital ridge [58]. We, therefore, analyzed published RNA-seq data from sorted primordial germ cells of staged female genital ridges between E11.5 to E18.5 [58]. Combining reads mapping to any of the 3 copies of *H2A.B*, we found a specific induction of *H2A.B* at the initiation of prophase of meiosis I, until entry into pachytene when it gradually decreased back to 0 (Fig 4C). This induction pattern mimics the meiotic transcriptional program (as indicated by *Prdm9*, Fig 4C) [58], although *H2A.B* is expressed at much lower levels than *Prdm9* in females and *H2A.B* in males [9,15,24]. Our preliminary analyses revealed no differences between WT and KO females in either synaptonemal

complex formation or H2A.X staining in pachytene cells isolated at E17.5 (S10 Fig). Thus, consistent with *H2A.B* being both a paternal- and a maternal-effect gene, we find that it is expressed during both male and female meiosis.

## Discussion

In this study, we generated and characterized a triple KO mutant of *H2A.B* in a *C57BL6/J* mouse genetic background. A recent report characterized a triple KO mouse in a different genetic background (*FVB/NJArc*) and found a significant albeit subtle effect of *H2A.B* loss on litter sizes and spermatogenesis [26]. Surprisingly, although we observe similar trends, we did not observe any significant impact on spermatogenesis, sperm, or testis morphology as a result of *H2A.B* loss. It is possible that these subtle differences are a result of different genetic backgrounds (*FVB/NJArc* versus *C57BL6/J*) or the conditions under which the different mice strains were raised or assayed.

Instead of an effect on spermatogenesis, we uncovered 3 novel functions of *H2A.B* by investigating the biological consequences of its loss in mice. First, CUT&RUN chromatin profiling with antibodies to specific histone H3 modifications revealed that loss of *H2A.B* leads to a global increase in the abundance of sub-nucleosomal fragments in haploid spermatids. Closer examination revealed that the sub-nucleosomal fragments do not correspond to specific genomic locations but rather represent a genome-wide pattern of partially unwrapped nucleosomes. This finding might initially appear surprising since H2A.B-containing nucleosomes have been shown to have smaller MNase footprints also suggestive of unwrapping [11,12,17]. However, the sub-nucleosomal fragments we have recovered are much smaller ($\leq$80 bp) than the footprints reported for H2A.B-containing nucleosomes (approximately 120 bp). Furthermore, corresponding to the loss of H2A.B nucleosomes, we found that mono-nucleosomes fragments released by CUT&RUN are indeed larger in KO than in WT chromatin (S5 Fig).

We hypothesize that these alterations in chromatin structure might reveal a role for H2A.B in histone-to-protamine exchange during spermiogenesis [59–61]. Previous studies using untethered MNase showed that transition structures occurring during histone-to-protamine exchange also produce sub-nucleosomal fragments of <120 bp in size [23,62–64]. Interestingly, the loss of another short H2A, *H2A.L.2*, prevents these changes from occurring in an orchestrated manner, ultimately leading to male sterility [23]. Since H2A.B deposition occurs earlier than H2A.L.2 in mice, we propose that H2A.B may serve as a checkpoint—delaying the premature disruption of nucleosomes by transition structures. The loss of H2A.B could therefore lead to a higher incidence of transition structures and may explain the higher than expected CMA3 staining (increased protamination) in sperm produced by KO males. The fact that these signatures of unwrapping are particularly prominent over H3K4me3 nucleosomes could also explain the previously reported effect of H2A.B on transcribed regions [24–26].

Our second novel finding is that *H2A.B* is expressed during female meiosis [58], and this expression has meaningful consequences on embryonic development. One possibility is that, as observed in males, *H2A.B* is deposited in female germ cells and functions to regulate the chromatin remodeling that occurs following fertilization [65,66]. Even though H2A.B is itself not found to be deposited in mouse sperm [21,42], a previous study found H2A.B staining in the first cleavage cycles of mouse zygotes [54]. This suggests the intriguing possibility that this detected H2A.B might have been inherited on oocyte genomes. Future in-depth chromatin profiling experiments in oocytes and early embryos can address these possibilities directly both for H2A.B and other short H2A histone variants, which are also believed to be exclusively "testis-specific."

The third and most significant finding from our study is the discovery that *H2A.B* is both a paternal-effect and a maternal-effect gene. This is the first instance of a histone variant with

this dual function. This novel function manifests both at the level of postimplantation embryo development and viability, but not preimplantation development or fertilization success. Although we did not observe differences in birth weights between pups produced from KO × KO and WT × WT crosses under laboratory conditions, we posit that differences in embryonic weight translate to an evolutionarily significant fitness defect in the wild [33,34,67,68]. Moreover, even under laboratory conditions, we observed a 4-fold increase in inviable embryos when both parents lack *H2A.B*, such that 1 in 10 embryos conceived in KO × KO crosses were inviable. Thus, we can conclude that *H2A.B* is required for optimal fitness in mice.

How does *H2A.B* exerts its parental effect? H2A.B seems to be lost after meiosis is completed, in elongating spermatids, and is not found in mature mouse sperm [21,24,42], although we cannot completely rule out the possibility that diminishingly small amounts of H2A.B might be present in mouse sperm. Thus, its postfertilization function must be exerted through other means than direct deposition in the zygote. *H2A.B*'s effect on splicing regulation or its proposed association with PIWI-related protein might affect the inheritance of specific RNAs required for optimal embryonic development and survival [25,69]. It is also possible that H2A.B alters the regulation of imprinted loci that regulate development through the parent-of-origin action of noncoding RNAs and DNA methylation [35]. Indeed, H2A.B has been shown to localize to a few imprint control regions in mouse embryonic stem cells [70]. However, a model that invokes paternal imprinting at specific loci cannot fully account for our findings that maternal *H2A.B* can overcome most but not all of the deleterious effects caused by loss of paternal *H2A.B*.

Instead, our finding of widespread rather than locus-specific alterations of chromatin structure in haploid male germ cells supports an alternative model whereby the deposition of *H2A.B* during meiosis globally prepares chromatin for proper postfertilization development. Such function has been previously described for chromatin regulators and histones in animals and plants [55,57,71–76]. For example, Drosophila HP1E is not deposited in sperm, yet the loss of HP1E leads to catastrophic defects in embryonic mitosis [71]. Consistent with this hypothesis, our CMA3 staining results suggest that mature sperm chromatin is altered upon H2A.B loss, potentially leading to increased protamination, which could have maladaptive consequences.

Our findings indicate that *H2A.B* is an important gene required for postimplantation embryonic development in mice, whose function can be redundantly supplied by either parent. However, this model cannot explain either the high rate of turnover of *H2A.B* including complete loss in different mammalian lineages or the rapid evolution of *H2A.B* coding sequences. Instead, *H2A.B*'s unusual evolution, together with its X chromosomal location, is highly suggestive of its involvement in a genetic conflict [9,28,40,77]. Although we find no evidence for *H2A.B* participation in X-versus-Y meiotic drive, our discovery of a role for *H2A.B*'s in embryonic development and viability suggests that it might participate in a different form of genetic conflicts, such as parental antagonism [38–40]. Parental antagonism arises when alleles have different fitness effects on progeny depending on whether they are inherited maternally or paternally. One classic form of parental antagonism occurs by genomic imprinting via DNA methylation, in which paternal and maternal alleles battle for resource allocation in embryos in utero [35]. In this model, paternally derived alleles encode "demand enhancers" that maximize maternal resource allocation to sired progeny, whereas maternally derived alleles encode "demand inhibitors" that distribute resources more equitably among all her progeny, enhancing her overall reproductive fitness [33,34].

*H2A.B* meets several but not all tenets of the "classic" parental antagonism model [33,34]. For example, paternal *H2A.B* presence ensures maximal fitness for all sired progeny, independent of maternal genotype. Notably, paternal *H2A.B* increases both fetal and placental weights, consistent with its role as a "demand-enhancer" [34,78]. Moreover, consistent with the antagonism model, defects arising from loss of *H2A.B* do not manifest during fertilization, or

preimplantation but rather during postimplantation embryonic development, exactly the stage where battles for resource allocation are predicted to occur in utero [33–35]. Furthermore, maternal *H2A.B* can eliminate fetal weight differences between progeny sired by WT or KO sires. Thereby, maternal H2A.B acts as capacitor allowing females to exert control over resource allocation to their progeny. Yet, maternal *H2A.B* does not act as a canonical "demand-inhibitor" as would be expected under the parental antagonism model. Instead, paternal and maternal *H2A.B* appear to have a synergistic rather than antagonistic effect on promoting embryo fitness; the most severe growth and viability phenotypes in utero are only seen in cases when neither parent encodes *H2A.B* (Fig 3A).

However, the paternal and maternal *H2A.B* contributions to embryonic viability are not symmetric. The most obvious indication of this comes from our long-term breeding assays (S9C Fig). Perinatal mortality (measured by number of litters with 1 or more dead pups) is nearly 30% in crosses between WT females and KO males, but only approximately 20% in KO × WT or KO × KO crosses, similar to WT × WT crosses. These findings imply that paternal *H2A.B* must counteract the negative influence of maternally contributed *H2A.B* on perinatal survival, explaining why mortality is higher in WT × KO crosses than in KO × KO crosses. This implies that the biparental contributions of *H2A.B* are not entirely redundant with each other, either due to their timing of action, or due to different mechanisms, or both. These different parent-of-origin effects of *H2A.B* on embryonic development and survival suggest that *H2A.B* has different male and female optima, consistent with a genetic conflict.

Their X-chromosomal linkage could further explain the rapid evolution and gene turnover of *H2A.B* genes in mammalian genomes under the parental antagonism model. Because of their X-linkage, any male-beneficial (e.g., demand-enhancer) allele of *H2A.B* would be fully acted upon by selection in hemizygous males, even as its female-deleterious effects could be masked in HET females [79]. As a result, novel alleles of *H2A.B* could recurrently arise and rise to high frequency in populations, followed by suppressor mutations that decrease female-detrimental effects to restore female fitness. Unexpectedly, our findings suggest that *H2A.B* alleles (or paralogs) themselves could evolve such a suppressor role. Thus, X-linkage and parental antagonism explain the rapid evolution and turnover of *H2A.B* alleles through repeated rounds of selection for optimizing male, followed by female-beneficial function. Moreover, H2A.B might be lost in mammalian species that relieve parental antagonism, for example, by sexual monogamy [80]. By revealing a novel role in postimplantation embryonic development, our study thus finds a satisfying explanation for why the evolutionary origins of *H2A.B* coincided with complete in utero development that evolved in placental mammals 150 million years ago [9].

## Materials and methods

### CRISPR editing of mouse zygotes

Two gRNAs were used for generating *H2A.B* KOs: gRNA 1: 5′gcctggtggaacagcatctg3′ and gRNA2: 5′cgcaagctccagcaacctgc3′. These sequences have perfect complementarity to all 3 intact *H2A.B* genes in the mouse genome, with only 1 mismatch in the most 5′ nucleotide of gRNA2 in the *H2afb3* paralog (C to G). Using PCR, we synthesized products containing the T7 promoter (5′ upper case in forward primer), the specific gRNA and a common trans-activating crispr RNA (tracrRNA) (upper case in 3′ and reverse primer):

gRNA1_Forward: CCTTAATACGACTCACTATAGGgcctggtggaacagcatctgGTTTTAGAGCT
    AGAAATAGC,

gRNA2_Forward: CCTTAATACGACTCACTATAGGcgcaagctccagcaacctgcGTTTTAGAGC
TAGAAATAGC, and

gRNA_Reverse_primer: AAAAGCACCGACTCGGTGCCACTTTTTCAAGTTGATAACG
GACTAGCCTTATTTTAACTTGCTATTTCTAGCTCTAAAAC.

The gRNAs were then in vitro transcribed using the MEGAshortscript T7 kit (Life Tech
AM1354, Thermo-Fisher, United States of America). gRNAs and Cas9 mRNAs were microinjected
into B6;SJLF1/J zygotes at a concentration of 75 ng/uL each gRNA and 100 ng/ul Cas9 mRNA.

## Triple KO line validation and genotyping

Founder animals, F(0), with putative mutations in any *H2A.B* paralog were mated to *C57BL/6J*
strain mice to assess germline transmission. F(1)s were sequenced using primers spanning the
whole *H2A.B* ORF and flanking regions:

H2afb3: B1_F:CAAGCCAAACTTTTCTTGAGGATGT, B1_R:TGAGCAGGTCAGCCAAGA
AG;

H2afb2: B1/2_F:CAGGGAAAACTGTCTTCGAGA, B2_R:GATACTACTAAAGAGAATCA
GCTTATAAAGAAGC;

H2afb1: B3_F:ACACGGAGAACTGTCTTCAAAG, B3_R:CAGAATCAAATACGGAAACAT
TTTTCCC.

Amplicons were cloned into pCR™4-TOPO TA vector (Thermo-Fisher) and verified by
Sanger sequencing (S2 Fig).

We used PCR genotyping with the same primer pairs to maintain our H2A.B triple KO
line. For mutant H2afb3 (B1), we used a forward primer specific to the first deletion (deletion1,
S2 Fig): B1_mut_F: TTGCTGTGAGCCTGGTGGAGG. This primer produces an amplicon in
the mutant (279bp) but not in the WT (S1B Fig). All pups were monitored for mutations in all
3 paralogs across all mating schemes throughout the duration of the project.

## SNP genotyping for C57BL6/J

To measure the percentage of C57BL6/J genetic background, SNP genotyping was performed
by DartMouse (Lebanon, New Hampshire, USA). High molecular weight DNA was prepared
from tissues collected postmortem on 1 hemizygous male and 1 homozygous sibling female
produced from backcross number 7. A total of 5,307 BL6/J specific SNPs were probed by Illu-
mina Infinium BeadChip (Illumina, USA).

## Fertility assay and breeding

Four WT and 6 KO sibling age-matched 8-week-old males (backcross 7) were allowed to mate
sequentially with individual C57BL/6J females (64 to 70 days, Charles River, USA) for 9
months. Once the female appeared pregnant, she was housed separately. If detected, vaginal
plugs were recorded. Live and dead progenies born from pregnant females were recorded
along with their sex and genotype. Long-term breeding experiments were performed by mat-
ing 1 male and 1 female (8 weeks to 12 weeks old, backcross 5 to 7). Each pair was allowed to
produce up to 5 consecutive litters. The litters were genotyped and sexed at weaning.

## Daily sperm production

Five pairs of WT and KO sibling males between 2 and 8 months old were used. Testes were
dissected, cleaned of fat, and weighed. A portion of 1 testis was dissected, fixed in modified

Davidson's fixative or Bouin's solution, and stained with H-PAS (Hematoxylin-Periodic Acid Schiff) for histological analysis. For testicular spermatid counts and daily sperm production (DSP), 15 to 40 mg of testis was individually weighed and homogenized in 0.1 M sodium phosphate buffer, pH 7.4, containing 0.1% Triton X-100, using 8 strokes in an all-glass Kontes 15-mL homogenizer. Ten microliters of the homogenate were assayed using a Neubauer phase contrast hemocytometer in duplicate. Total homogenization-resistant spermatids per gram and per organ and DSP were calculated by correcting for the number of squares counted, dilution, volume, weight, and duration of homogenization-resistant spermatids [81,82], as previously described [83,84].

## Sperm motility and morphology

For motility analyses, spermatozoa were collected from the caudae epididymides from the right side of each animal. We placed the isolated cauda into modified Tyrodes medium with 1% bovine serum albumin (Sigma #9647, Sigma-Aldrich, USA), cut the tubule about 6 times with fine microscissors, and allowed sperm to flow out into the medium. The medium was collected, and large pieces were allowed to settle. The suspended spermatozoa were diluted in the same medium as required, and 6 μL was placed into a Leja fixed coverslip slide for observation in a phase contrast microscope at 200× magnification. The percentage of sperm with beating tails was interpreted as % motile.

For morphology analyses, caudae epididymidal sperm from 4 pairs of WT and KO males were smeared, and 100 sperm cells were subjected to a multiple anomaly morphology analysis at 1000×. Cells were counted as either "normal," when no defect was observed or with "abnormal heads," "bent necks," or "tails defects" (as shown in S7D Fig).

## Sperm DNA damage assay

Epididymes were collected from pairs of sibling WT and KO males at 3, 4, and 5 months old. Caudal epidydimal sperm were allowed to swim out in 1× PBS and snap frozen in liquid $N_2$ at a final concentration of approximately $10^7$ cells/mL in the dark. Neutral comet and alkaline diffusion assays were performed as previously described in detail [85]. For single-cell electrophoresis comet assays, sperm nuclear DNA was categorized as undamaged if it formed a ragged circle, moderately damaged if there were strands extending up to the diameter of undamaged DNA, and highly damaged if the electrophoresed DNA extended farther. For alkaline diffusion, the percentage of apoptotic cells with diffuse DNA and a hazy outline were calculated from a total of 100 cells. Necrotic cells were counted at the same time; they have expanded but well-delineated nuclear DNA. Nuclei of cells that were alive at the time of processing were distinguished from the other 2 patterns by their compact and bright DNA staining without a halo or diffusion.

## CMA3 staining

Epididymes were collected in Hank's Buffered Salt Solution from 2 pairs of sibling WT and KO males at 3 months old. Caudal epididymal sperm were allowed to swim out in solution and smeared onto polylysine-coated slides. Slides were then fixed by immersion in 1:1 ethanol: acetone and dried for at least 10 min. CMA3 staining was performed as previously described [86]. Briefly, each slide was incubated with 100 μl of CMA3 solution (0.25 mg/mL CMA3 (Sigma-Aldrich), 10 mM $MgCl_{2+}$ in Mcilvaine buffer (Boston BioProducts, USA)) at room temperature in the dark for 20 min. Slides were rinsed once in 1× PBS, air-dried for at least 20 min, and mounted in 15 μL of Vectashield with DAPI (Vector Laboratory, USA).

## TUNEL assay

Paraffin-embedded testis cross sections were melted for 45 min at 65°C and deparaffinized in Histoclear twice for 5 min. Slides were then subject to a series of ethanol washes (100%, 90%, 70%, and deionized water) for 3 min each. Slides were transferred to 1× PBS and subjected to the in situ cell death detection kit (Sigma-Aldrich) according to manufacturer's instructions. Briefly, slides were proteinase K–treated for 20 min at 37°C in humid chamber, washed, and incubated with TUNEL reagent for 1 h at 37°C in a humid chamber.

## Germ cell spreads

Germ cell spreads were performed as previously described [87] except for the following modifications: for males, both testes were obtained from humanely killed adult males postweaning age (2 pairs of 8-week-old WT and KO siblings); for females, fetal ovaries were dissected from E17.5 embryos in 1× PBS. Ovaries were placed in hypo-extraction buffer for 12 min at room temperature. We performed co-staining with primary antibodies that were diluted in 1X antibody dilution buffer (ADB): SCP3 (ab15093, Abcam, USA) at 1:200, MLH1 (550838, BD Pharmingen, USA) at 1:60 and H2A.X (ab2893, Abcam, USA) at 1:200. Antibody staining was performed in a humid chamber at 37°C overnight. The next day, the rubber cement was removed, and the slides were soaked in 1X ADB and then washed 2 × 30 min in 1X ADB. Staining with secondary antibodies (Alexa 488 anti-rabbit (A21206, Thermo-Fisher, USA), 1:500, and Alexa 568 anti-mouse (A11031, Thermo-Fisher, USA), 1:400) was performed in 1× ADB in a humid chamber at 37°C for 2 h. Slides were then washed for 3 × 30 min in 1X PBS and mounted in Vectashield with DAPI (Vector Laboratory, USA). For female samples, slides were washed 1 × 20 min after primary staining and washed for 1 × 30 min in 1× PBS after secondary staining.

## Testis cross section and pathology

The testes and epididymides of 3 pairs of KO and WT males (2 to 3 months old) were dissected, fixed and embedded in paraffin, and sectioned at the Fred Hutchinson Cancer Research Center (FHCRC) Histopathology Core. Tissue composition, structure and count of meiotic and somatic cells of the seminiferous tubule, as well as overall pathology (presence of germ cells in the lumen, damage of tubules, multinucleated giant cells, apoptotic Sertoli cells) were assessed using haemotoxylin and eosin (H&E) staining.

## In vitro fertilization and Imaging

For in vitro fertilization studies, 26 WT females (8 weeks old, C57BL/6J) were superovulated by Inhibin (i.p., KYD-010-EX-X5; CosmoBio, USA) followed by hCG (i.p., 5 IU, Sigma, Cat# 230734), 48 h later. Cumulus–oocyte complexes were collected and in vitro fertilized with sperm from WT or H2A.B KO males (3 replicates, oocytes collected from 4 females per replicate per genotype) as previously described [88]. The total number of oocytes per replicates was distributed as follows: WT1 = 126; WT2 = 134; WT3 = 108; KO1 = 144; KO2 = 141, KO3 = 137. Fertilized oocytes were incubated in KSOM at 37°C, 5% $O_2$, 5% $CO_2$ until 4 days post-insemination and analyzed for progressive development.

All imaging was performed on a Zeiss epifluorescence Axiovert 200M (ZEISS, Germany) inverted microscope.

## Cell sorting

Live cell sorting was performed as previously described [43]. Briefly, testes were dissected and the tunica were removed. The testes were dissociated in 1 mL of 1× dissociation buffer

(Collagenase A (5 mg/mL), Dispase (5 mg/mL), DNaseI (0.5 mg/mL) in PBS) for 45 min at 37˚C under shaking. Cells were then counted and stained for 90 min at 37˚C with Hoechst 33342 dye (1 mg/mL, Sigma-Aldrich) at with 5 μL per $10^6$ cells in DMEM 10% FBS. Cells were then washed in PBS and resuspended with Propidium Iodide (P1304MP, Thermo-Fisher, USA) for cell sorting on a BD FACSAria II (BD Bioscience, USA). Haploid cells were collected based on DNA content and red UV shift according to previously published protocols [43]. Cells were collected in complete cell culture media, kept on ice, and processed within 4 h for AutoCUT&RUN.

## Antibodies for AutoCUT&RUN

We used Rabbit anti-H3K4me3 (1:100, Active Motif Cat#39159), rabbit anti-H3K27me3 (1:100, 9733S, Cell Signaling Tech, USA), and rabbit anti-H3K9me3 (1:100, Abcam, ab8898). Rabbit anti-mouse IgG (1:100, Abcam, USA, ab46540) was used as a negative control.

## AutoCUT&RUN, sequencing, and mapping

For AutoCUT&RUN analyses, 2 WT and KO non-mated sibling pairs (2 months old) were used for replicate experiments. AutoCUT&RUN was performed according to the method previously outlined [44]. Detailed, updated protocols are available at protocols.io (dx.doi.org/10.17504/protocols.io.ufeetje). Up to 24 barcoded AutoCUT&RUN libraries were pooled per lane at equimolar concentration for paired-end 25 × 25 bp sequencing on a 2-lane flow cell on the Illumina HiSeq 2500 (Illumina, USA) platform at FHCRC Genomics Shared Resource. We aligned paired end reads to the mm10 mouse genome assembly (Genome Reference Consortium) using Bowtie2 version 2.2.5 [89] with options: local—very-sensitive-local—no-unal—no-mixed—no-discordant—phred33 -I 10 -X 700. To generate normalized counts in bedGraph format, total mapped fragments counts were normalized to genome-wide coverage. Representative UCSC tracks can be seen in S11 Fig.

## Data analysis

BAM files were processed using bedTools [90] to generate BED files corresponding to mapped fragments and UCSC bedGraphToBigWig programs to generate bigwig files. PCR duplicates were removed using MarkDuplicates from Picard Tools ("Picard Toolkit." 2019, Broad Institute, USA). Enriched chromatin domains, or peaks, were called using sparse enrichment analysis for CUT&RUN (SEACR) with an IgG file as a control with the stringent setting (S3 Table) [91]. For each histone mark, the total peaks called was produced by combining all peaks from WT and KO samples. FRiPs were generated with bedtools coverage. For fragment downsampling (to compare identical total counts in WT and KO samples), we used the command line tool "shuf". Nucleosome fuzziness scores were called in KO or WT samples using the software DANPOS2 with default settings [92]. Heatmaps were produced using normalized bigWigs with deepTools2 [93] implemented into the public Galaxy server at usegalaxy.org [53].

## Embryo collection and weighing

Pregnant dams were euthanized 14 days following observation of a reproductive plug. Embryos and placentas were dissected and washed with 1X PBS. For each conceptus, tissues were collected into 1.75 mL microcentrifuge tubes and kept on ice until weighing on a fine scale (XS105 Dual Range, Mettler Toldeo, USA). Upon dissection, embryos were scored as "severely delayed" if they were properly implanted but displayed no visible vascularization and delayed organ development compared to neighboring E14.5 conceptuses with stereotypical

development. Implantation sites with necrotic embryonic or placental structures were scored as resorbed.

## Ethics statement

Animal husbandry and experimentation was reviewed, approved, and monitored under the Institutional Animal Care and Use Committee at Fred Hutchinson Cancer Research Center (Protocol ID: PROTO000050927). Protocols used for the breeding and tissue collection of *H2A.B* KO animals were performed at the Fred Hutchinson Comparative Medicine animal facility. The facility is fully accredited by the Association for Assessment and Accreditation of Laboratory Animal Care and complies with all United States Department of Agriculture, Public Health Service, Washington State, and local animal welfare regulations.

## Supporting information

**S1 Fig. Generation of H2A.B triple KOs.** (A) Schematic of the CRISPR/Cas9 strategy used to produce *ΔH2A.B* animals. All *H2afb* coding loci mapped in mm10 are shown with their coordinates in parentheses. Details on the sequenced deletions for each paralog are shown at the bottom. Scissors indicate the regions of CRISPR/Cas9-induced cuts. Arrows indicates the location of genotyping primers (see Materials and methods for details) (B) A PCR genotyping example of *ΔH2A.B* alleles from pups of a HET × WT mating. A 1-kb plus ladder is shown on the left side of each gel. For *H2afb1* and *H2afb2*, primers amplify a (smaller) KO or (larger) WT fragment. *H2afb3* primers are specific for the KO allele and do not amplify a band in the WT. HET, heterozygous; KO, knockout; WT, wild-type.
(TIFF)

**S2 Fig. Validation of H2A.B triple KOs.** Alignment of *H2afb* paralog ORFs with Sanger traces from the sequencing of *H2afb1* (A), *H2afb2* (B), and *H3afb3* (C) from ΔH2A.B animals. Annotated are ORFs (yellow boxes) gRNAs (purple arrow) and deletions. gRNA, guide RNA; KO, knockout; ORF, open reading frame.
(TIFF)

**S3 Fig. SNP genotyping results comparing *ΔH2A.B* animals with pure *C57BL/6J* reference strain.** Percentages of HET, homozygous, or non-*C57BL/6J* SNPs in pure WT, H2A.B triple KO males or females are shown (Materials and methods). HET, heterozygous; KO, knockout; WT, wild-type.
(TIFF)

**S4 Fig. Profiling the chromatin features of haploid germ cells.** (A) Schematic of the flow cytometry (scatter plot) and MNase-based profiling (red box) used in this study. (B, C, D) AutoCUT&RUN normalized fragment counts (blue dots) over all peaks called for the histone modifications H3K9me3 (A), H3K27me3 (B), and H3K4me3 (C). KO counts are shown on the y-axis and WT on the x-axis. Linear regression and $R^2$ values are shown in red. (E) Percentage of total mapped fragment assigned to chromosome X (blue), Y (orange), or autosomes (blue) (see S5 Data). Y-axis is in log10 scale. Results for fragments released by AutoCUT&RUN with IgG, H3K27me3, H3K4me3, and H3K9me3 are shown for both KO and WT samples. KO, knockout; MNase, micrococcal nuclease; WT, wild-type.
(TIFF)

**S5 Fig. Features of the fragments released by CUT&RUN in haploid spermatids.** Schematic of fragment size (x-axis) distributions (y-axis) in WT and KO spermatids (replotted from Fig 1A). Sub-nucleosomal fragments below 120 bp are thought to correspond to partially

unwrapped nucleosomes. In WT (blue), most nucleosomal fragments (above 120 bp) have the size of H2A.B containing mono-nucleosomes. In *H2A.B KO* (red), nucleosomal fragments have the size of H2A containing mono-nucleosomes. Note that H2A.B nucleosomes have a smaller MNase footprints than H2A nucleosomes. MNase cleavage is indicated by arrowheads. KO, knockout; MNase, micrococcal nuclease; WT, wild-type.
(TIFF)

**S6 Fig. Regional distribution of AutoCUT&RUN fragments.** (A, B, C) Heatmaps of normalized counts (blue scale) of short fragments (<121 bp) and large fragments (>149 bp) over the top 2,000 low fuzziness nucleosomes in WT or KO samples are shown. Regions are centered on nucleosomes profiled with H3K9me3 (A), H3K27me3 (B), and H3K4me3 (C). (D) Distribution of fragments over the TSS of all + strand RefSeq genes. (E) KO/WT ratios of small/large fragment ratios in peaks defined by H3K27me3, H3K9me3, and H3K4me3 (see S6 Data). Log10 (ratios) for replicate 1 and replicate 2 are shown on the x- and y-axes, respectively. $R^2$ values are shown in all 3 cases. KO, knockout; TSS, transcriptional start site; WT, wild-type.
(TIFF)

**S7 Fig. Testes and sperm function of *H2A.B* KO males.** (A) Representative images of testis cross sections of WT and KO males (see Materials and methods) (B) Mean (with standard deviation, gray bars) DSP and (C) percentages of motile sperm measured from the same WT (blue) and KO (red) males are shown. (D) Fractions of sperm with normal morphology, abnormal heads, bent necks, or tails defects measured in WT or H2A.B KO males are indicated. Means and standard deviations are shown across replicates (WT $n = 4$; KO $n = 3$). Example images of each morphology are shown on the right. Supporting information can be found as S7 Data. DSP, daily sperm production; KO, knockout; WT, wild-type.
(TIFF)

**S8 Fig. Analysis of DNA damage and CMA3 staining of *H2A.B* KO male germ cells.** (A) Fraction of seminiferous tubules with at least 1 TUNEL positive cell in WT (blue) or KO (red) testis cross sections. *n* is the number of tubules counted. (B) Crossover events measured using staining against the proteins SCP3 and MLH1 (see Materials and methods) in germ cell spreads from WT (blue) and KO (red) animals. Mean (with standard deviation) number of chromosomes per pachytene cell with 0 (E0), 1 (E1), 2 (E2), or 3 (E3) cross overs are indicated. "n" is the total number of chromosomes counted for each genotype. (C) Fraction of gammaH2A.X positive cells at pachytene or late-pachytene in WT (blue) and KO (red) germ cell spreads. (D) Proportion (%) of epididymal sperm cells showing damage by Comet or alkaline diffusion (Alkaline) assays in WT (blue) or KO (red) samples. Individual replicates of WT and KO sibling pairs are shown separately (Rep1, 2, 3, 4, 5). (E) Fraction of CMA3 positive sperm cells in WT (blue) or KO (red) samples. Two replicates are shown. Two-hundred sperm were counted for each. Representative images of CMA3 positive cells are shown on the right. Supporting information can be found as S8 Data. CMA3, chromomycin A3; KO, knockout; MLH1, MutL homolog 1; SCP3, synaptonemal complex protein 3; TUNEL, terminal deoxynucleotidyl transferase dUTP nick end labeling; WT, wild-type.
(TIFF)

**S9 Fig. Sex ratios and mortality.** (A, B) Punnett squares of matings between WT females and WT males (top) or KO males (bottom) in fertility assays (A) or long-term breeding (B). The ratio of female pups (right squares) over male pups (lefts squares) denote the sex ratios. $X^{KO}$ denotes *H2A.B KO* chrX. (C) Percentage of litters with at least 1 dead pup in long-term breeding (see S9 Data). Mating is indicated as dam × sire. The total number of litters is shown inside

each bar. chrX, X chromosome; KO, knockout; WT, wild-type.
(TIFF)

**S10 Fig. Preliminary assessment of pachytene cells in *H2A.B KO* females.** (A) Representative images of WT (left panel) and KO (right panel) pachytene cells, from E17.5 germ cell spreads (see Materials and methods), stained for gammaH2A.X (green) and SCP3 (red). (B) Percentage of gammaH2A.X positive pachytene cells in WT (blue) and KO (red) females (see S8 Data) and total number of cells counted are shown inside each bar. KO, knockout; SCP3, synaptonemal complex protein 3; WT, wild-type.
(TIFF)

**S11 Fig. UCSC browser track of AutoCUT&RUN data.** A segment of chr7 is shown, with Refseq genes, and transposon insertions (black bars). Normalized fragment counts across 2 replicates is shown for each AutoCUT&RUN dataset. WT samples (blue tracks) and KO samples (red tracks) are plotted for H3K27me3, H3K4me3, H3K9me3, and IgG. KO, knockout; WT, wild-type.
(TIFF)

**S1 Table. Fragment size analysis of AutoCUT&RUN.** Total number of fragments of sizes <121 bp and >149 bp from equal sampling of total fragments in WT and KO. Chi-squared significance of observed KO counts versus expected counts in WT. KO, knockout; WT, wild-type.
(XLSX)

**S2 Table. Fraction of total fragments in peaks for small <121 bp and large >149 bp fragments.**
(XLSX)

**S3 Table. Total number of peaks called for each histone mark by SEACR.** SEACR, sparse enrichment analysis for CUT&RUN.
(XLSX)

**S4 Table. Results of male fertility assays.** Four WT and 6 KO male siblings were bred sequentially to a total of 28 and 38 females, respectively. Detailed statistics and significance of statistical tests comparing KO to WT are shown. KO, knockout; WT, wild-type.
(XLSX)

**S1 Data. Fragment counts used in Fig 1A–1C.**
(XLSX)

**S2 Data. Individual litter sizes used in Fig 2A and raw number of embryos used in Fig 2C.**
(XLSX)

**S3 Data. Numbers of resorbed or severely delayed embryos used in Fig 3A and individual values of embryo weights used in Fig 3B.**
(XLSX)

**S4 Data. Values of embryos weight used in Fig 4A and replicate RPKM values in Fig 4C.** RPKM, reads per kb per million.
(XLSX)

**S5 Data. Fraction of read counts mapping to chrX, chrY, and autosomes used in S4E Fig.** chrX, X chromosome; chrY, Y chromosome.
(XLSX)

**S6 Data. Fragment counts in peaks and ratios used in S6E Fig.**
(XLSX)

**S7 Data. Replicates values for DPS, motility, and morphology used in S7A, S7B, and S7D Fig.** DPS, daily sperm production.
(XLSX)

**S8 Data. Cell counts in TUNEL assay (S8A Fig), chromosome crossover counts (S8B Fig), H2A.X positive male meiotic cells (S8C Fig), Replicate sperm DNA damage (S8D Fig), CMA3 positive sperm cells (S8E Fig), and H2A.X positive female pachytne cells (S10B Fig).** TUNEL, terminal deoxynucleotidyl transferase dUTP nick end labeling.
(XLSX)

**S9 Data. Number of litters with at least 1 dead pup in long term breeding assays used in S9C Fig.**
(XLSX)

## Acknowledgments

We thank current and past members of the Henikoff and Malik labs for helpful discussions throughout the duration of this project. We especially thank Smitha Pillai and Amanda Koehne for help with histopathology analyses, the Fred Hutchinson Comparative Medicine Animal Facility for invaluable advice and help with mouse husbandry, and Caiying Guo and her team from the Gene Targeting & Transgenic Facility at the Janelia Research campus for the CRISPR KO service. We also thank Ching-Ho Chang, Ines Drinnenberg, Kevin Forsberg, Rini Kasinathan, Tera Levin, Mia Levine, Pravrutha Raman, Courtney Schroeder, Janet Young, and Sarah Zanders for their comments on the manuscript.

## Author Contributions

**Conceptualization:** Antoine Molaro, Anna J. Wood, Steven Henikoff, Harmit S. Malik.

**Data curation:** Antoine Molaro, Anna J. Wood, Steven Henikoff, Harmit S. Malik.

**Formal analysis:** Anna J. Wood, Derek Janssens, Charles H. Muller, Steven Henikoff, Harmit S. Malik.

**Funding acquisition:** Antoine Molaro, Anna J. Wood, Harmit S. Malik.

**Investigation:** Antoine Molaro, Anna J. Wood, Derek Janssens, Selina M. Kindelay, Michael T. Eickbush, Steven Wu, Priti Singh, Charles H. Muller, Harmit S. Malik.

**Methodology:** Antoine Molaro, Anna J. Wood, Derek Janssens, Selina M. Kindelay, Michael T. Eickbush, Steven Wu, Priti Singh, Charles H. Muller.

**Project administration:** Antoine Molaro, Anna J. Wood, Steven Henikoff, Harmit S. Malik.

**Resources:** Derek Janssens, Steven Wu, Harmit S. Malik.

**Software:** Steven Wu.

**Supervision:** Antoine Molaro, Steven Henikoff, Harmit S. Malik.

**Validation:** Antoine Molaro, Anna J. Wood, Charles H. Muller, Steven Henikoff, Harmit S. Malik.

**Visualization:** Antoine Molaro, Anna J. Wood, Harmit S. Malik.

**Writing – original draft:** Antoine Molaro, Anna J. Wood, Harmit S. Malik.

**Writing – review & editing:** Antoine Molaro, Anna J. Wood, Derek Janssens, Charles H. Muller, Steven Henikoff, Harmit S. Malik.

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
