## [Editor Report · Decision Letter 0]

28 Sep 2020

Dear Harmit, 

Thank you for submitting your manuscript entitled "Biparental contributions of the H2A.B histone variant control embryonic development in mice" for consideration as a Research Article by PLOS Biology.

The revision of your manuscript from Review Commons has now been evaluated by the PLOS Biology editorial staff as well as by an academic editor with relevant expertise and I am writing to let you know that we would like to send your submission back to the original reviewers.

Please re-submit your manuscript within two working days, i.e. by Sep 30 2020 11:59PM.

Best wishes,

Ines

--

Ines Alvarez-Garcia, PhD,

Senior Editor

PLOS Biology

---

## [Decision Letter · Decision Letter 1]

22 Oct 2020

Dear Harmit,

Thank you for submitting your revised Research Article from Review Commons entitled "Biparental contributions of the H2A.B histone variant control embryonic development in mice" for publication in PLOS Biology. I have now obtained advice from the three original reviewers and have discussed their comments with the Academic Editor.

We're delighted to let you know that we're now editorially satisfied with your manuscript. The reviews are attached below and you will see that they are all very positive without further requests. Nevertheless, Reviewer 2 would like you to add some discussion of a potential mechanistic explanation for the embryonic roles of H2A.B, but we will leave it up to you to decide.

I would also like to encourage you to submit one of the pachytene cells images shown in Fig. S10 as potential cover image.

Before we can formally accept your paper and consider it "in press", we also need to ensure that your article conforms to our guidelines. A member of our team will be in touch shortly with a set of requests. As we can't proceed until these requirements are met, your swift response will help prevent delays to publication. Please also make sure to address the data and other policy-related requests noted at the end of this email.

- a cover letter that should detail your responses to any editorial requests, if applicable

*Copyediting*

*Published Peer Review History*

*Early Version*

Best wishes,

Ines

--

Ines Alvarez-Garcia, PhD,

Senior Editor,

ialvarez-garcia@plos.org,

PLOS Biology

ETHICS STATEMENT:

-- Thanks for sending us the details of the animal care licence. Please also include an approval number.

DATA POLICY:

Fig. 1A-C; Fig. 2A-D; Fig. 3A, B; Fig. 4A, C; Fig. S4E; Fig. S5; Fig. S6E; Fig. S7B-D; Fig. S8A-E; Fig. S9C and Fig. S10B

Also, please make sure that the the raw CUT&RUN data that has been deposited at GEO is accessible at the time of acceptance.

Reviewers’ comments

Rev. 1:

In the revised version of their manuscript, the authors have satisfyingly addresses the issues/questions that I had raised in my first review. The study is novel and highly significant; I support its publication without additional changes.

NB. I detected two typos: one in Fig S4E title, the other Line 492 (missing word in sentence "Our second novel finding is that H2A.B expressed during female meiosis").

And while re-reading my review I realize I wrote "CUT&TAG" instead of Cut&Run, my apologies for the confusion!

Rev. 2:

This is a significantly improved version of the manuscript previously submitted to Review Commons. The authors have successfully addressed most of my comments.

This is a highly relevant work, giving an important contribution for a better understanding of the developmental roles of short H2A histone variants in placental mammals.

Given its relevance to the field, I recommend this manuscript in its current format for publication in PLoS Biology. Yet, the lack of a precise mechanistic explanation for the embryonic roles of H2A.B, should nevertheless be considered before a final editorial decision.

Rev. 3:

I went through the revised manuscript and feel that the manuscript is now acceptable.

---

## [Editor Report · Decision Letter 2]

30 Nov 2020

Dear Dr Malik,

On behalf of my colleagues and the Academic Editor, Laurence D Hurst, I am pleased to inform you that we will be delighted to publish your Research Article in PLOS Biology. 

PRODUCTION PROCESS

Before publication you will see the copyedited word document (within 5 business days) and a PDF proof shortly after that. The copyeditor will be in touch shortly before sending you the copyedited Word document. We will make some revisions at copyediting stage to conform to our general style, and for clarification. When you receive this version you should check and revise it very carefully, including figures, tables, references, and supporting information, because corrections at the next stage (proofs) will be strictly limited to (1) errors in author names or affiliations, (2) errors of scientific fact that would cause misunderstandings to readers, and (3) printer's (introduced) errors. Please return the copyedited file within 2 business days in order to ensure timely delivery of the PDF proof. 

If you are likely to be away when either this document or the proof is sent, please ensure we have contact information of a second person, as we will need you to respond quickly at each point. Given the disruptions resulting from the ongoing COVID-19 pandemic, there may be delays in the production process. We apologise in advance for any inconvenience caused and will do our best to minimize impact as far as possible.

EARLY VERSION

PRESS 

Kind regards,

Erin O'Loughlin

Publishing Editor, 

PLOS Biology

on behalf of

Ines Alvarez-Garcia,

Senior Editor

PLOS Biology